# Movement Artifact Direction Estimation Based on Signal Processing Analysis of Single-Frame Images

**DOI:** 10.3390/s25247487

**Published:** 2025-12-09

**Authors:** Woottichai Nonsakhoo, Saiyan Saiyod

**Affiliations:** Hardware-Human Interface and Communications Laboratory (H2I-Comm Lab), Department of Computer Science, College of Computing, Khon Kaen University, Khon Kaen 40002, Thailand; nonsakhoo@cassia.kku.ac.th

**Keywords:** motion blur, movement artifact, direction estimation, signal processing, single-frame image, self-similarity analysis, impulse response, multiplicative noise, noise detection, direction detection, noise assessment

## Abstract

Movement artifact direction and magnitude are critical parameters in noise detection and image analysis, especially for single-frame images where temporal information is unavailable. This paper introduces the Movement Artifact Direction Estimation (MADE) algorithm, a signal processing-based approach that performs 3D geometric analysis to estimate both the direction (in degrees) and weighted quantity (in pixels) of movement artifacts. Motivated by computational challenges in medical image quality assessment systems such as LUIAS, this work investigates directional multiplicative noise characterization using controlled experimental conditions with optical camera imaging. The MADE algorithm operates on multi-directional quantification outputs from a preprocessing pipeline—MAPE, ROPE, and MAQ. The methodology is designed for computational efficiency and instantaneous processing, providing interpretable outputs. Experimental results using precision-controlled apparatus demonstrate robust estimation of movement artifact direction and magnitude across a range of image shapes and velocities, with principal outputs aligning closely to ground truth parameters. The proposed MADE algorithm offers a methodological proof of concept for movement artifact analysis in single-frame images, emphasizing both directional accuracy and quantitative assessment under controlled imaging conditions.

## 1. Introduction

Motion blur is a prevalent artifact in images caused by camera movement or subject motion during exposure, significantly degrading image quality and making content analysis challenging. In medical imaging, this phenomenon, also known in as movement artifact, occurs across photography, computer vision, and medical imaging applications, resulting in blurred images that lack sharpness and detail, leading to potential misinterpretation of image content [1,2,3]. Movement artifacts present significant challenges across diverse imaging modalities, from Magnetic Resonance Imaging (MRI) procedures where patient movement during scanning causes ghosting and blurring artifacts [4,5], to endoscopic ultrasonography where motion affects vessel detection capabilities [6], and ultrasound pulse-echo techniques where tracking moving objects requires sophisticated signal processing [7,8,9]. The impact of motion blur extends to clinical applications such as colonoscopy where blurry frames from motion, out-of-focus conditions, and water jets can lead to missed detections [10], and medical imaging quality assessment where movement artifacts significantly affect evaluation accuracy [11]. Specialized measurement techniques have been developed for various applications, including photoacoustic velocimetry for blood flow measurement in microvasculature using cross-correlation of successive signal pairs [12] and super-resolution ultrasound imaging with two-stage motion estimation methods combining affine and nonrigid estimation [13]. Accurately estimating the direction of motion blur is crucial for effective image restoration and analysis, as the directional information provides valuable insights about camera or subject movement, enabling improved image quality enhancement and supporting advanced image analysis tasks [14,15,16,17,18]. The assessment of motion blur quality has been advanced through the development of objective blur level metrics using point spread function analysis [19] and structural similarity indices that degrade with movement artifacts [20].

Deep learning approaches have gained significant attention for motion blur detection and estimation in recent years. Convolutional neural networks (CNNs) have shown promising results in identifying blur patterns and estimating blur parameters [21,22,23,24,25]. Notable implementations include Motion Correction-Net (MC-Net) for motion compensation in brain MRI achieving approximately 40 ms processing times [21], end-to-end networks for rotary motion deblurring addressing polar coordinate transformation errors [24], and two-stage deblurring modules using high frequency residual learning [25]. Advanced architectures encompass classification models for directional analysis [26], autoencoder frameworks for blind deconvolution [27], attention mechanisms, generative adversarial networks, and transformer-based approaches [28,29]. Despite achieving high accuracy in controlled scenarios, these methods face significant limitations: (1) requirement for large annotated datasets, (2) high computational complexity during training and inference with GPU dependency, (3) limited interpretability of learned features, and (4) domain-specific performance that may not generalize across imaging conditions [30], rendering them impractical for real-time or resource-constrained applications.

Multi-frame approaches leverage temporal information from sequential image acquisitions to estimate motion blur direction and magnitude, analyzing frame-to-frame variations, optical flow patterns, and temporal gradients to infer motion characteristics [31,32]. Several studies have explored these sequential methods, including optical flow-based tracking [33] and multi-frame joint detection approaches [34], but these approaches share the common limitation of requiring multiple frames for analysis. Advanced motion correction techniques have been developed for medical imaging applications, including elastic motion correction in PET imaging through mass preservation optical flow algorithms [35] and cone-beam computed tomography movement artifact correction using morphological analysis of blood vessels [36]. The literature reveals a progression from simple frame differencing to sophisticated optical flow estimation and multi-object tracking systems [37,38], with applications extending to moving object detection using enhanced optical flow estimation and feature threshold segmentation [39]. Frequency domain methods utilize Fourier transforms and spectral analysis to identify motion blur characteristics, typically assuming that lower frequencies correspond to motion blur while higher frequencies represent sharpness and edges [40,41]. Cortés-Osorio et al. [16] introduced discrete cosine transform (DCT) for velocity estimation from single motion blurred images, achieving mean absolute errors of 4.66° in direction estimation. Advanced frequency domain approaches include reorganized DCT representation for autofocus applications [42], frequency-Bessel transform methods for extracting multi-order dispersion curves from background noise [43], and time-domain inverse digital recursive methods for characteristic frequency signal removal [44]. However, the fundamental limitation of frequency domain methods is the loss of precise spatial localization information. While these techniques can identify the presence and general characteristics of motion blur, they cannot pinpoint the exact location and direction of blur artifacts in the spatial domain [45,46], limiting their effectiveness for targeted correction and analysis.

Single-frame images present particularly challenging scenarios for motion blur analysis, as they lack temporal information that multi-frame approaches rely upon. The need for low-latency processing capabilities in applications such as medical imaging, surveillance systems, and computer vision further emphasizes the importance of computationally efficient solutions for single-frame directional estimation [47,48,49,50,51]. Theoretical foundations for modern signal processing approaches provide essential frameworks for addressing these challenges through comprehensive methodologies encompassing stochastic signals, parameter estimation, and adaptive filtering techniques [52].

Iterative approaches apply multiple noise reduction techniques and measure signal-to-noise ratios (SNRs), selecting optimal results through comparative analysis [53,54]. These methods often combine spatial and temporal information, repeatedly applying different filtering strategies until achieving satisfactory SNR metrics, but this iterative nature makes them costly and time-consuming. Time-domain methodologies for motion assessment have emerged as alternatives to traditional approaches, offering advantages in specific applications where temporal analysis provides superior motion characterization [55]. Advanced signal processing techniques for time-of-flight and frequency estimation have shown promise in reducing background noise interference and improving parameter extraction accuracy [56], with comprehensive reviews of measurement techniques emphasizing the importance of fluctuating characteristics in dynamic systems [57]. Specialized system approaches have been developed for specific applications, including SABER (Systems Approach to Blur Estimation and Reduction) for X-ray radiographs using point spread function modeling and numerical optimization [58], and QP-based visual servoing schemes that limit motion blur through velocity constraints and image gradient sharpness metrics [59].

Traditional spatial domain methods focus on direct analysis of pixel intensities, gradients, and local features to identify blur characteristics, including edge detection, gradient analysis, and local variance computations [60,61]. Recent studies have explored robust estimation procedures and adaptive filtering techniques for motion parameter identification [45,46], while specialized applications have demonstrated movement artifact elimination in optical coherence tomography angiograms [62] and controllable motion-blur effects for post-processing applications [63]. Advanced spatial domain techniques include convolved feature vector based adaptive fuzzy filters for impulse noise removal [64], super-resolution algorithms with outline feature extraction for small object detection [65], and precision coordinate alignment methods using motion capture systems [66]. However, most existing spatial methods lack sophisticated geometric analysis capabilities for precise directional estimation. Current spatial domain approaches generally lack comprehensive geometric frameworks for motion blur direction estimation, and few approaches integrate advanced geometric analysis with signal processing principles to achieve precise directional localization in 3D space representations.

Despite these advances, several critical gaps remain in the literature that limit the effectiveness of existing motion blur direction estimation methods. First, most frequency-domain approaches lack the ability to provide precise directional information in the spatial domain [67], as they cannot pinpoint the exact location and orientation of motion blur artifacts [40,46,68,69]. While these methods may detect the presence of motion blur, they often fail to deliver the spatial localization and directional vectors necessary for targeted correction, which is especially problematic in applications requiring geometric analysis of blur patterns. Second, the computational complexity of current methods—particularly deep learning and frequency-domain techniques—poses significant challenges for low-latency or large-scale applications, as they demand substantial resources and processing time [21,22]. Third, many existing approaches are domain-specific and rely on assumptions about noise characteristics or motion patterns that may not generalize across different image types or acquisition conditions.

The need for robust directional multiplicative noise characterization is particularly evident in AI-based image analysis systems where classifier performance depends on image quality. In systems such as the Liver Ultrasound Image Analysis System (LUIAS) for cholangiocarcinoma surveillance system [14,47], multiple factors including acoustic noise, probe angle selection, and acquisition conditions affect periductal fibrosis detection and classification robustness. While many degradation factors have established mitigation pathways, multiplicative noise in single static images remains challenging to characterize computationally. Multiplicative noise captures degradations that scale with signal intensity—such as motion-induced patterns—which cannot be addressed through conventional additive noise reduction alone. This motivates developing signal processing frameworks that quantify such patterns under controlled conditions, providing methodological foundations for future domain-specific investigations. The present work addresses this through optical imaging experiments where ground-truth parameters are precisely controlled, enabling rigorous validation without ultrasound physics confounds. Critically, precise directional and magnitude estimates would enable future compensation strategies that could systematically enhance periductal fibrosis detection and classification performance in such systems. Figure 1 illustrates the relationship between this methodological work and LUIAS motivation, where system infrastructure components are highlighted in gray, recent research developments are indicated in blue, and the green block with dotted ellipse delineates the specific scope addressed in this paper.

A critical research gap remains: existing methods lack computationally efficient frameworks for characterizing multiplicative noise in single-frame images under controlled conditions with known ground-truth parameters. While frequency-domain and deep learning approaches offer sophisticated analysis, they sacrifice either spatial precision or computational tractability. Moreover, developing efficient movement artifact compensation strategies fundamentally requires precise direction and magnitude estimates—yet no prior work provides both parameters simultaneously through computationally tractable signal processing approaches validated under controlled conditions where ground-truth is known.

We propose the Movement Artifact Direction Estimation (MADE) framework—a signal processing approach using controlled optical imaging to characterize directional multiplicative noise through mathematical modeling and empirical validation with precision-controlled apparatus. The MADE algorithm performs 3D geometric analysis on multi-directional quantification outputs from a preprocessing pipeline (MAPE, ROPE, MAQ), extracting directional features across four axes and transforming them into precise directional estimates. This methodology delivers spatial domain analysis, low computational complexity, and comprehensive multi-directional capability validated through controlled experiments. This work addresses the estimation component, providing the directional and magnitude parameters that future compensation algorithms would require as inputs. While motivated by LUIAS computational challenges (Figure 1), this study establishes fundamental signal processing methodologies using optical imaging, with artifact compensation strategies and domain-specific applicability reserved for future work.

## 2. Materials and Methods

This section presents our signal processing framework for movement artifact direction estimation in single-frame images. The methodology combines a detection and quantification pipeline (MAPE, ROPE, MAQ) with the MADE algorithm for 3D geometric analysis. The pipeline quantifies movement artifacts across four principal directions, while MADE transforms these outputs into precise directional estimates through optimal triplet selection, center of mass calculation, and weighted distance computation. Figure 2 illustrates the methodology overview, with blue components representing previous work and green components indicating novel contributions.

### 2.1. Signal-Domain Representation of Movement Artifacts

This section establishes the mathematical foundation for modeling movement artifacts from a signal processing perspective, distinct from conventional pixel-based image processing approaches and ultrasound acoustic beamforming. The proposed model captures both rotational and translational components of motion, providing a rigorous analytical basis for subsequent algorithmic development and empirical validation.

In signal processing theory, the observed signal is represented as a combination of the original signal s(t), multiplicative noise $n_{m}\left(t\right)$, and additive noise $n_{a}\left(t\right)$:(1)$$y\left(t\right)=s\left(t\right)\cdot n_{m}\left(t\right)+n_{a}\left(t\right).$$ For movement artifact analysis, emphasis is placed on the multiplicative component $n_{m}\left(t\right)$, where motion introduces structured spatial distortions rather than stochastic interference. The transformation from the spatial domain to the signal domain enables deterministic modeling of movement-induced modulation as a function of motion parameters.

Let $I:R^{2}\to R$ denote the spatial image function and T:I↦s(t) represent the spatial-to-signal transformation operator. The movement artifact manifests as a multiplicative modulation term $n_{m}\left(t;\varphi \right)$ parameterized by the orientational angle φ∈[0,2π), yielding the observed signal:(2)$$y\left(t;\varphi \right)=s\left(t\right)\cdot n_{m}\left(t;\varphi \right).$$ Here, φ serves as the fundamental control parameter governing the multiplicative structure, enabling directional characterization of motion-induced distortions through signal-domain analysis.

Within this framework, the object’s motion is described by a continuous-time parametric trajectory that combines rotational and translational components:(3)$$r\left(t;\varphi \right)=r_{r}\left(t\right)+r_{t}\left(t;\varphi \right)=\left(A\cos(\omega t),\;A\sin(\omega t)\right)-vt\left(\cos\varphi ,\;\sin\varphi \right),$$
where A>0 is the oscillation amplitude, ω>0 is the angular frequency, v≥0 is the linear velocity, t∈[0,T] is the exposure time, and φ defines the motion orientation. The rotational term $r_{r}\left(t\right)$ induces cyclic displacement, while the translational drift $r_{t}\left(t;\varphi \right)$ produces monotonic elongation aligned with φ. For conceptual illustration, see Figure 3a.

The image formation process is governed by the integration of a spatial point spread function (PSF) $h:R^{2}\to R_{\geq 0}$ along the parametric trajectory r(t;φ). The PSF *h* is assumed normalized ($\int_{R^{2}}h\left(u,v\right)\;du\;dv=1$) and compactly supported, representing the system’s response to a point source. Typical choices include the Gaussian kernel,(4)$$h\left(x,y\right)=\frac{1}{2\pi \sigma^{2}}\exp\left(-\frac{x^{2}+y^{2}}{2\sigma^{2}}\right),$$
where σ>0 is the standard deviation controlling the spread of the blur, and $\left(x,y\right)\in R^{2}$ are spatial coordinates. For an idealized point response, the Kronecker delta is used:(5)h(x,y)=δ(x)δ(y),
where δ(·) denotes the Dirac delta function, representing an impulse at the origin. Physically, *h* describes how light or signal from a single point is distributed in space, controlling the degree of sharpness or blur in the resulting image. The observed image I(x,y;φ) is generated by accumulating the PSF along the trajectory r(t;φ), effectively forming a superposition of the PSF at each position visited during the motion:(6)$$I\left(x,y;\varphi \right)=\int_{0}^{T}h\left(x-x(t;\varphi ),\;y-y(t;\varphi )\right)\;dt.$$ This formulation can be equivalently expressed as a convolution of the PSF *h* with a trajectory-induced measure $\mu_{\varphi }$:(7)$$I\left(\cdot ;\varphi \right)=h*\mu_{\varphi },$$
where $\mu_{\varphi }$ is a measure (e.g., Dirac or distribution) supported along the path traced by r(t;φ), assigning mass to each location visited during exposure. This links the physical motion model directly to the observed image structure.

To quantify the directional extent of blur, we define the arc-length projection as:(8)L(φ)=∫0Tdrdt(t;φ)dt(9)=∫0TA2ω2+v2−2Aωvsin(ωt)cosφ+cos(ωt)sinφdt
where *A* represents the amplitude of oscillatory motion, ω is the angular frequency, and *v* denotes the constant velocity component. The variable *t* is time, while φ specifies the direction along which the blur is measured. The integrand computes the instantaneous speed along the trajectory, combining both oscillatory and linear motion effects.

This explicit expression demonstrates how the orientational angle φ modulates the total directional path elongation, linking the underlying motion parameters to the observed blur characteristics in the image.

For practical simulation, the above integrals are approximated by a Riemann sum with step size s>0: $t_{n}=ns$, n=0,1,…,N−1, N=⌊T/s⌋. The discrete trajectory samples are(10)$$\left(x_{n},y_{n}\right)=r\left(t_{n};\varphi \right),\;I\left[i,j;\varphi \right]\approx \sum_{n=0}^{N-1}K\left(i-y_{n},\;j-x_{n}\right),$$
where *K* is a discrete Kronecker or Gaussian kernel, and the approximation error is O(s) under standard smoothness of r. The mapping $S_{s}$ yields $I_{s}=S_{s}\left[h,r\left(\cdot ;\varphi \right)\right]$. All non-angular parameters (A,ω,v,T,h,s) affect scale or smoothness, but only φ determines the orientation of the drift and thus the anisotropy of the resulting blur.

The directional blur pattern is a deterministic functional of φ, i.e., I=F(φ) for fixed parameters. However, the inverse problem—estimating θ from a single observed frame *I*—is non-trivial due to the entanglement of rotational and drift components and the absence of temporal context. This motivates our proposed estimation pipeline, which leverages multi-directional statistical features (MAPE, ROPE, MAQ) and geometric optimization (MADE) to approximate the inverse mapping $F^{-1}:I\mapsto \theta$.

The φ is the known ground-truth or fundamental orientation parameter that determines the direction of the movement artifact, while θ denotes the estimated direction obtained by our methodology. The φ governs the orientation of the drift, which in turn shapes the anisotropic spatial patterns observed in the image. Our methodology systematically extracts and quantifies these patterns through a multi-stage pipeline, culminating in a robust estimation of the underlying movement artifact direction θ. For visual interpretation, Figure 3b illustrates the 2D projection of the modeled movement artifact, while Figure 3c presents the corresponding empirical image, demonstrating how the theoretical framework translates to practical image analysis.

To clarify the problem statement and the principal parameters estimated by our approach, Figure 4 illustrates a unified circular object moving across a uniform background, captured as it traverses from right to left in front of the camera at a velocity $v_{j}\approx 0.2\;m/s$. The motion occurs at angles $\varphi \approx 30^{\circ }$ and $210^{\circ }$, with ground truth labels available for every $15^{\circ }$ increment from $0^{\circ }$ to $165^{\circ }$ and their complementary angles, enabling comprehensive $360^{\circ }$ analysis. Both velocity *v* and orientation φ are known during ground truthing and serve as reference labels for evaluating the proposed method. The figure highlights the two principal parameters estimated—direction θ and magnitude $q_{i}$—as well as the outputs of the Detection and Quantification Pipeline ($P_{m}^{\mathrm{loc}}$ and ${\hat{g}}_{i}$), which are adaptively incorporated as required by the procedure. Solid lines and parameters denote primary analysis, while faint lines and parameters represent secondary or complementary analysis.

### 2.2. Data Preparation

The data preparation stage primarily employs empirical acquisition of single-frame images exhibiting controlled movement artifacts, with mathematical synthesis serving to confirm theoretical hypotheses. This approach ensures that the proposed methodology is grounded in real-world imaging conditions while maintaining analytical rigor. For empirical validation, we constructed a real-world dataset using a precision-controlled apparatus that induces motion in a unified black object on white background across 12 orientation configurations, spanning orientation angles $\varphi_{i}\in \left\{15^{\circ }k|k=0,1,2,\ldots ,11\right\}$ with uniform angular spacing $\Delta \varphi =15^{\circ }$. Each primary orientation $\varphi_{i}$ inherently defines its corresponding complementary orientation $\varphi_{i}+180^{\circ }$, establishing bidirectional symmetry pairs $\left(\varphi_{i},\varphi_{i}+180^{\circ }\right)$ that comprehensively cover all possible orientations within the full angular domain $\left[0^{\circ },360^{\circ }\right)$. For each orientation configuration, the apparatus generates motion at 11 predefined velocities $v_{j}\in \left\{0.1j|j=0,1,2,\ldots ,10\right\}\;m/s$ with uniform velocity spacing Δv=0.1m/s, resulting in 12×11=132 distinct parameter combinations. For each parameter combination $\left(\varphi_{i},v_{j}\right)$, multiple image acquisitions are performed with repetition count $n_{i,j}\in \left[1,10\right]\subset N$, yielding a comprehensive dataset cardinality of $\sum_{i=0}^{11}\sum_{j=0}^{10}n_{i,j}$ images, with theoretical bounds 12×11×[1,10]=[132,1320] total image instances. The apparatus consists of a velocity control subsystem (a geared DC motor rotating a metallic rod with an object, allowing fine-grained speed control) and a shutter triggering subsystem (a microcontroller triggers the camera when the object interrupts a laser beam, ensuring consistent capture timing). Each acquired image is annotated with its ground truth velocity (*v*) and orientation (φ), providing precise reference data for validation. Multiple images are acquired for each velocity–orientation combination, and all images are post-processed for spatial consistency by cropping and alignment to center the object and standardize the region of interest. This unified approach, combining mathematical modeling and empirical acquisition with comprehensive orientation and velocity coverage, ensures that the dataset is both physically meaningful and analytically tractable, providing a robust foundation for subsequent movement artifact quantification and analysis.

### 2.3. Detection and Quantification Pipeline

The detection and quantification pipeline is designed to extract robust, multi-directional features that quantify movement artifacts in single-frame images. This stage consists of three core algorithms—MAPE, ROPE, and MAQ—each grounded in classical signal processing and tailored for computational efficiency and interpretability.

Movement Artifact Position Estimation (MAPE) systematically detects and localizes movement artifacts by analyzing the spatial structure of normalized signals along each scanline. For a normalized image matrix $I_{\mathrm{norm}}\in R^{M\times N}$, each row *m* is treated as a one-dimensional signal $x_{m}=\left[I_{\mathrm{norm}}\left(m,1\right),\ldots ,I_{\mathrm{norm}}\left(m,N\right)\right]$. The algorithm computes the normalized impulse response (self-similarity profile) for each scanline:(11)am[l]=∑n=1Nxm[n]xm[n−l],l=−(N−1),…,N−1(12)γm[l]=∑n=1Nxm[n]2·∑n=1Nxm[n−l]2(13)rm[l]=am[l]γm[l],l=−(N−1),…,N−1

Peaks in $r_{m}\left[l\right]$ are detected using a prominence criterion, which ensures that only structurally significant features are retained. Prominence is defined as the vertical distance between a peak and its lowest contour line, i.e., the minimum value to which the signal must descend before rising to a higher peak.

Let $L_{m}$ denote the set of lag indices *l* where $r_{m}\left[l\right]$ attains a local maximum that satisfies the prominence criterion. The lag position of the second most prominent peak for each scanline *m* is then defined as:(14)$$\begin{array}{c} P_{m}^{\mathrm{loc}}={arg max}_{l\in L_{m}\setminus \left\{0\right\}}\left(\mathrm{prominence}(r_{m}\left[l\right])\right) \end{array}$$
where arg max returns the lag index of the second most prominent peak (excluding l=0, which corresponds to the main peak at zero lag). Here, the notation P (for “second peak”) with subscript “loc” (for “location”) is used to denote the set of detected second peak locations. The set of all detected artifact positions across all scanlines is $P^{\mathrm{loc}}={⋃}_{m=1}^{M}\left\{P_{m}^{\mathrm{loc}}\right\}$, and a symmetry-based lag complementation is applied to account for both displacement directions, yielding the complementary set ${\bar{P}}^{\mathrm{loc}}=\left\{-l|l\in P^{\mathrm{loc}}\right\}$. The resulting sets $P^{\mathrm{loc}}$ and ${\bar{P}}^{\mathrm{loc}}$ succinctly encode artifact peak positions and their complements, forming the basis for quantification. For all sums involving $x_{m}\left[n-l\right]$, we define $x_{m}\left[k\right]=0$ whenever k<1 or k>N (i.e., zero-padding is used for out-of-bounds indices).

ROPE estimates the undistorted reference origin for each scanline, providing a stable anchor for artifact quantification. Each scanline is smoothed using an exponential moving average (EMA) with parameter α:(15)EMAi,1=xi,1(16)EMAi,j=αxi,j+(1−α)EMAi,j−1,j=2,…,N The first difference (slope) of the EMA sequence is computed to highlight structural transitions:(17)$$s_{i,j}={\mathrm{EMA}}_{i,j+1}-{\mathrm{EMA}}_{i,j}$$ The dominant transition for each scanline *i* is identified as the index ${\hat{g}}_{i}$ where the slope attains its maximum absolute value:(18)$${\hat{g}}_{i}={arg max}_{j}\left|s_{i,j}\right|$$ This transition is retained only if its magnitude exceeds a threshold $T=\tau \cdot \max_{i}\left|{\hat{v}}_{i}\right|$, where τ is a tunable parameter and ${\hat{v}}_{i}$ is the maximum slope for scanline *i*. The set of reference origins is then defined as(19)$$G=\left\{{\hat{g}}_{i}\;|\;|s_{i,{\hat{g}}_{i}}|>T\right\}$$
where G captures the most significant structural boundaries for each scanline, providing robust reference points for subsequent artifact quantification.

MAQ quantifies the spatial displacement between detected artifact positions and their reference origins, incorporating adaptive filtering to suppress outliers. For each scanline, the MAQ value is defined as the spatial distance between the reference origin ${\hat{g}}_{i}$ and the closest valid artifact position (from either $P_{i}^{\mathrm{loc}}$ or ${\bar{P}}_{i}^{\mathrm{loc}}$), subject to a plausibility threshold τ:(20)$$q_{i}=\left\{\begin{array}{cc} {\hat{g}}_{i}-{\bar{P}}_{i}^{\mathrm{loc}}, & \mathrm{if}\;P_{i}^{\mathrm{loc}}\;\mathrm{is}\;\mathrm{NaN}\;\mathrm{and}\;{\bar{P}}_{i}^{\mathrm{loc}}\;\mathrm{is}\;\mathrm{defined}\\ {\hat{g}}_{i}-P_{i}^{\mathrm{loc}}, & \mathrm{if}\;P_{i}^{\mathrm{loc}}\;\mathrm{is}\;\mathrm{defined}\;\mathrm{and}\;{\bar{P}}_{i}^{\mathrm{loc}}\;\mathrm{is}\;\mathrm{NaN}\\ {\hat{g}}_{i}-P_{i}^{\mathrm{loc}}, & \mathrm{if}\;|{\hat{g}}_{i}-P_{i}^{\mathrm{loc}}\left|\leq \right|{\hat{g}}_{i}-{\bar{P}}_{i}^{\mathrm{loc}}|\\ {\hat{g}}_{i}-{\bar{P}}_{i}^{\mathrm{loc}}, & \mathrm{otherwise} \end{array}\right.$$
with $|q_{i}|>\tau$ resulting in suppression ($q_{i}=\mathrm{NaN}$). The final set $Q=\{q_{i}|i=1,2,\ldots ,M\}$ provides a robust, scanline-wise quantification of movement artifact magnitude.

This detection and quantification pipeline yields a compact, multi-directional summary of movement artifacts, with each step mathematically grounded and computationally efficient. The outputs—artifact positions, reference origins, and quantified displacements—serve as critical inputs for the subsequent geometric analysis in the MADE algorithm, enabling precise and interpretable motion blur direction estimation.

### 2.4. MADE Algorithm

The MADE algorithm represents the core innovation of our methodology, transforming the directional MAQ sets Q from the detection and quantification pipeline into precise directional estimates through 3D geometric analysis. The algorithm employs a systematic four-stage approach: directional MAQ preparation, 3D space mapping with reliability prioritization, optimal triplet selection, and direction estimation with weighted distance calculation. This geometric framework transcends traditional signal processing limitations by integrating spatial relationships and statistical reliability measures to achieve unprecedented accuracy in single-frame motion blur direction estimation.

#### 2.4.1. Directional MAQ Preparation

The MADE algorithm begins by applying the detection and quantification pipeline to four directional components: horizontal, vertical, diagonal, and antidiagonal. Each direction requires a specific enumeration strategy to preserve geometric structure and enable optimal feature extraction without interpolation artifacts.

For a given centered cropped image $I_{\mathrm{crop}}\in R^{M\times N}$, directional matrices $D_{d}$ are constructed as follows:

Horizontal and Vertical Directions: These correspond directly to image rows and columns, respectively. The horizontal matrix uses the original image structure ($D_{\mathrm{horizontal}}=I_{\mathrm{crop}}$), while the vertical matrix applies a 90° rotation ($D_{\mathrm{vertical}}=R_{90^{\circ }}\left(I_{\mathrm{crop}}\right)$). Diagonal and Antidiagonal Directions: These require more complex processing involving element collection along diagonal lines, center alignment, and spatial cropping to maintain consistency. The diagonal direction collects elements where i+j=k, while the antidiagonal direction uses the constraint j−i+M=k. Detailed formulations for these transformations, including centering offsets, cropping boundaries, and reflection operations, are provided in Appendix A, with the illustration shown in Figure 5.

After matrix construction, row-wise normalization is applied to ensure consistent intensity scaling:(21)$$I_{\mathrm{norm}}\left(k,:\right)=\frac{D_{d}\left(k,:\right)-\mu_{k}}{\sigma_{k}}$$
where $\mu_{k}$ and $\sigma_{k}$ are the row mean and standard deviation, respectively (see Section A.1 for complete definitions).

Each normalized directional matrix $I_{\mathrm{norm}}$ is processed through the detection and quantification pipeline (MAPE, ROPE, MAQ) to yield directional MAQ sets:(22)$$Q^{*}=\left\{\mathrm{MAQ}\left(I_{\mathrm{norm}}\left(k,:\right)\right)|k=1,\ldots ,\dim_{1}\left(I_{\mathrm{norm}}\right)\right\}$$

With all four directional MAQ sets computed and spatially aligned, the algorithm proceeds to 3D space mapping for integrated geometric analysis and robust direction estimation.

#### 2.4.2. Three-Dimensional Space Mapping

In this stage, the MADE algorithm integrates the directional MAQ statistics to construct a geometric representation in 3D space. For each direction *d*, let $K_{d}$ denote the number of lines in the *d*-th direction of $Q^{*}$. The mean and standard deviation of the MAQ values for direction *d* are defined as(23)$${\bar{q}}_{d}=\frac{1}{K_{d}}\sum_{k=1}^{K_{d}}Q_{d,k}^{*},\;\sigma_{d}=\sqrt{\frac{1}{K_{d}}\sum_{k=1}^{K_{d}}{\left(Q_{d,k}^{*}-{\bar{q}}_{d}\right)}^{2}}$$
where $Q_{d,k}^{*}$ is the MAQ value for the *k*-th line in direction *d*.

To enhance the influence of variability and to allow for tunable sensitivity in the geometric mapping, the standard deviation for each direction is scaled by a configurable boost factor β>0:(24)$$\sigma_{d}^{\prime }=\beta \cdot \sigma_{d}$$
where β is a user-defined parameter. This boosted standard deviation $\sigma_{d}^{\prime }$ is used as a measure of directional uncertainty or variability, and its role as the third dimension in the 3D mapping allows the algorithm to emphasize or de-emphasize the effect of dispersion in the quantification process. A higher β increases the impact of variability, while a lower β reduces it, providing flexibility for different application requirements.

To prioritize directions with greater reliability, a value swapping strategy is applied to the set $\left\{\sigma_{d}^{\prime }\right\}$. The rationale is that a lower standard deviation indicates more consistent (and thus more reliable) MAQ measurements in that direction. Therefore, the smallest value in $\left\{\sigma_{d}^{\prime }\right\}$ is assigned the highest reliability score, and the largest is assigned the lowest. This is achieved by swapping the smallest and largest values, as well as the two intermediate values, effectively inverting the order of significance. Let $\left\{\sigma_{\left(1\right)}^{\prime },\sigma_{\left(2\right)}^{\prime },\sigma_{\left(3\right)}^{\prime },\sigma_{\left(4\right)}^{\prime }\right\}$ be the ordered set such that $\sigma_{\left(1\right)}^{\prime }\leq \sigma_{\left(2\right)}^{\prime }\leq \sigma_{\left(3\right)}^{\prime }\leq \sigma_{\left(4\right)}^{\prime }$. The reliability scores $\left\{\sigma_{d}^{*}\right\}$ are then defined by(25)$$\sigma_{d_{1}}^{*}=\sigma_{\left(4\right)}^{\prime },\;\sigma_{d_{2}}^{*}=\sigma_{\left(3\right)}^{\prime },\;\sigma_{d_{3}}^{*}=\sigma_{\left(2\right)}^{\prime },\;\sigma_{d_{4}}^{*}=\sigma_{\left(1\right)}^{\prime }$$
where $d_{1},d_{2},d_{3},d_{4}$ are the directions corresponding to the original order of $\left\{\sigma_{d}^{\prime }\right\}$. This inversion ensures that directions with lower variability are given higher reliability scores, which will be reflected in the geometric analysis.

Each direction and its complement are then mapped to a point in 3D space, with coordinates determined by the mean MAQ value and the corresponding reliability score. Let $\left(x_{0},y_{0}\right)$ denote the image center (or origin). The eight 3D points are defined as(26)$$\begin{array}{cc} p_{1} & =[x_{0}+{\bar{q}}_{\mathrm{horizontal}},\;y_{0},\;\sigma_{\mathrm{horizontal}}^{*}]\\ p_{2} & =[x_{0}+{\bar{q}}_{\mathrm{diagonal}},\;y_{0}+{\bar{q}}_{\mathrm{diagonal}},\;\sigma_{\mathrm{diagonal}}^{*}]\\ p_{3} & =[x_{0},\;y_{0}+{\bar{q}}_{\mathrm{vertical}},\;\sigma_{\mathrm{vertical}}^{*}]\\ p_{4} & =[x_{0}-{\bar{q}}_{\mathrm{antidiagonal}},\;y_{0}+{\bar{q}}_{\mathrm{antidiagonal}},\;\sigma_{\mathrm{antidiagonal}}^{*}]\\ p_{5} & =[x_{0}-{\bar{q}}_{\mathrm{horizontal}},\;y_{0},\;\sigma_{\mathrm{horizontal}}^{*}]\\ p_{6} & =[x_{0}-{\bar{q}}_{\mathrm{diagonal}},\;y_{0}-{\bar{q}}_{\mathrm{diagonal}},\;\sigma_{\mathrm{diagonal}}^{*}]\\ p_{7} & =[x_{0},\;y_{0}-{\bar{q}}_{\mathrm{vertical}},\;\sigma_{\mathrm{vertical}}^{*}]\\ p_{8} & =[x_{0}+{\bar{q}}_{\mathrm{antidiagonal}},\;y_{0}-{\bar{q}}_{\mathrm{antidiagonal}},\;\sigma_{\mathrm{antidiagonal}}^{*}] \end{array}$$
where $\sigma_{d}^{*}$ is the reliability score for direction *d* after the value swapping operation.

This coordinate system [x,y,z] encodes the directional MAQ mean as spatial displacement in the *x* and *y* axes, and the reliability score as the *z* axis, as illustrated in Figure 6a. The resulting 3D geometric configuration yields a set of points $P\in R^{n\times 3}$, which serves as the input for the subsequent optimal triplet selection process. Briefly, the next optimal triplet selection section determines the three consecutive adjacent points in 3D space that maximize the total distance from the origin, ensuring both adjacency constraints and geometric significance for directional estimation. This step is critical for accurately identifying the predominant movement artifact direction by selecting the most representative triplet from the eight available points, as illustrated in Figure 6b.

#### 2.4.3. Optimal Triplet Selection

Given the points matrix $P\in R^{n\times 3}$ where n=8 represents the total number of directional points, and the origin point $O={\left[0,0,0\right]}^{T}$, the algorithm first calculates the Euclidean distance from the origin to each point:(27)$$d_{i}={∥P_{i}-O∥}_{2}=\sqrt{{\left(P_{i,x}-O_{x}\right)}^{2}+{\left(P_{i,y}-O_{y}\right)}^{2}+{\left(P_{i,z}-O_{z}\right)}^{2}}$$
where $P_{i}={\left[P_{i,x},P_{i,y},P_{i,z}\right]}^{T}$ represents the *i*-th point coordinates and $d_{i}$ is the corresponding distance from the origin.

To identify the optimal triplet, the selection process enumerates all possible consecutive adjacent triplets with wrap-around connectivity. For each triplet starting at index *i*, the three consecutive points are defined as:(28)$$\begin{array}{c} {\mathrm{idx}}_{1}=i,\;{\mathrm{idx}}_{2}=\mathrm{mod}\left(i,n\right)+1,\;{\mathrm{idx}}_{3}=\mathrm{mod}\left(i+1,n\right)+1 \end{array}$$ For each candidate triplet, the total distance is computed as the sum of individual distances:(29)$$D_{\mathrm{triplet}}\left(i\right)=d_{{\mathrm{idx}}_{1}}+d_{{\mathrm{idx}}_{2}}+d_{{\mathrm{idx}}_{3}}$$ The objective is to maximize this total distance while maintaining the adjacency constraint:(30)$$i^{*}=\arg\max_{i=1}^{n}D_{\mathrm{triplet}}\left(i\right)$$ Accordingly, the optimal triplet is defined as the three consecutive points $\left\{{\hat{p}}_{1},{\hat{p}}_{2},{\hat{p}}_{3}\right\}$ corresponding to the indices $\left\{{\mathrm{idx}}_{1}^{*},{\mathrm{idx}}_{2}^{*},{\mathrm{idx}}_{3}^{*}\right\}$ determined by $i^{*}$:(31)$$\begin{array}{c} {\hat{p}}_{1}=P_{i^{*}},\;{\hat{p}}_{2}=P_{\mathrm{mod}(i^{*},n)+1},\;{\hat{p}}_{3}=P_{\mathrm{mod}(i^{*}+1,n)+1} \end{array}$$

For consistency in subsequent analysis, the optimal triplet $\left\{{\hat{p}}_{1},{\hat{p}}_{2},{\hat{p}}_{3}\right\}$ is directly utilized in the center of mass calculation described in Section 2.4.4.

This selection approach ensures that the identified triplet maximizes the collective distance from the origin while preserving spatial adjacency, which is critical for maintaining geometric coherence in the subsequent center of mass computation. The adjacency constraint prevents the selection of noncontiguous points, thereby supporting a robust and interpretable direction estimation process.

#### 2.4.4. Direction Estimation

The direction estimation process constitutes the principal outcome of the MADE algorithm, yielding the estimated direction angle of the movement artifact in degrees. This step transforms the optimal triplet selection results into a definitive directional estimate that serves as the foundation for subsequent calculations, including the estimation of the actual MAQ distance in pixels.

The direction estimation process begins with the computation of the 3D center of mass from the optimal triplet $\hat{p}$ obtained from the previous stage. The center of mass represents the geometric centroid of the three most significant points in 3D space and serves as a representative point indicating the overall directionality of the movement artifact. The 3D center of mass is calculated as the arithmetic mean of the three optimal triplet coordinates:(32)$$C_{\mathrm{mass}}=\frac{1}{3}\sum_{i=1}^{3}p_{i}=\frac{1}{3}\left({\hat{p}}_{1}+{\hat{p}}_{2}+{\hat{p}}_{3}\right)$$
where $C_{\mathrm{mass}}={\left[C_{x},C_{y},C_{z}\right]}^{T}$ represents the center of mass coordinates in 3D space.

The primary direction vector is established from the origin to the center of mass, creating a directional vector that encapsulates the predominant movement artifact orientation. This vector is then projected onto the 2D plane (x-y coordinates) for angular representation, as the directional estimation focuses on the spatial plane where the movement artifacts manifest.

The direction angle is computed using the four-quadrant inverse tangent function to ensure proper angular representation across all quadrants:(33)$$\theta =\arctan\left(\frac{C_{y}}{C_{x}}\right)+\Phi_{\mathrm{quadrant}}$$
where $\Phi_{\mathrm{quadrant}}$ is the quadrant correction factor defined as:(34)$$\Phi_{\mathrm{quadrant}}=\left\{\begin{array}{cc} 0^{\circ } & \mathrm{if}\;C_{x}>0\\ 180^{\circ } & \mathrm{if}\;C_{x}<0\\ 90^{\circ } & \mathrm{if}\;C_{x}=0\;\mathrm{and}\;C_{y}>0\\ 270^{\circ } & \mathrm{if}\;C_{x}=0\;\mathrm{and}\;C_{y}<0 \end{array}\right.$$

This formulation ensures correct angular determination across all four quadrants of the coordinate system, avoiding the inherent ambiguity of the standard arctangent function. To ensure the angle representation falls within the standard range $\left[0^{\circ },360^{\circ }\right)$, a normalization process is applied:(35)$$\theta =\theta_{\mathrm{normalized}}=\left\{\begin{array}{cc} \theta & \mathrm{if}\;\theta \geq 0^{\circ }\\ \theta +360^{\circ } & \mathrm{if}\;\theta <0^{\circ } \end{array}\right.$$ Additionally, the complementary angle is calculated to provide a complete directional reference:(36)$$\theta^{\prime }=\left(\theta_{\mathrm{normalized}}+180^{\circ }\right)\;\mathrm{mod}\;360^{\circ }$$

Figure 7 illustrates the center of mass calculation for the optimal triplet, highlighting the geometric relationship between the estimated direction and its distance from the origin. The blue dot marks the center of mass, with the blue dashed line indicating its spatial displacement. Figure 7a provides a 3D perspective, while Figure 7b presents the corresponding 2D (top-down) view. The bidirectional blue dashed line with arrowheads denotes the estimated movement artifact direction, which serves as the principal output of the proposed methodology and forms the basis for subsequent weighted distance calculation.

#### 2.4.5. Weighted Distance Calculation

The weighted distance calculation refines the estimated movement artifact magnitude by aligning it with the principal direction θ determined in the previous stage. While directional MAQ values quantify artifacts along predefined axes, this step computes the true magnitude in pixels along the actual artifact orientation. The process constructs a circular adjacency matrix connecting each 3D point to its neighbor, forming an 8-sided polygon boundary around the origin. The intersection of the direction vector with the corresponding polygon edge yields the weighted distance, representing the artifact’s true strength along its principal direction. Figure 8 illustrates the geometric relationships and key steps in this procedure.

The neighbor connectivity matrix $N\in R^{n\times 6}$ is constructed such that each row *i* contains the coordinates of point $P_{i}$ and its circular neighbor $P_{i+1}$:(37)$$N_{i}=\left[P_{i},P_{i+1}\right]=\left[P_{i,x},P_{i,y},P_{i,z},P_{i+1,x},P_{i+1,y},P_{i+1,z}\right]$$
where the neighbor index follows circular adjacency, i.e., next_idx =(imodn)+1, ensuring that the last point connects back to the first and creates a closed polygon boundary. The resulting neighbor connectivity matrix establishes line segments that form the boundary of an 8-sided polygon in 3D space. Each row represents a directed edge connecting consecutive points, with the general structure:(38)$$N=\left[\begin{array}{cccccc} P_{1,x} & P_{1,y} & P_{1,z} & P_{2,x} & P_{2,y} & P_{2,z}\\ P_{2,x} & P_{2,y} & P_{2,z} & P_{3,x} & P_{3,y} & P_{3,z}\\ P_{3,x} & P_{3,y} & P_{3,z} & P_{4,x} & P_{4,y} & P_{4,z}\\ \vdots & \vdots & \vdots & \vdots & \vdots & \vdots \\ P_{n,x} & P_{n,y} & P_{n,z} & P_{1,x} & P_{1,y} & P_{1,z} \end{array}\right]$$
where each row *i* contains the start point coordinates $\left[P_{i,x},P_{i,y},P_{i,z}\right]$ followed by the end point coordinates $\left[P_{i+1,x},P_{i+1,y},P_{i+1,z}\right]$, with circular indexing ensuring that the final row connects back to the first point. This matrix representation enables systematic identification of polygon edges for subsequent ray-line intersection calculations, where each row defines a line segment through its start and end coordinates in 3D space.

The algorithm performs angular segmentation by dividing the 360° space into 8 equal segments of 45° each, corresponding to the polygon edges. The line segment identification process determines which polygon edge contains the center of mass angle θ:(39)$$LineSegmentIndex=\iota =\left\lfloor \frac{\theta }{\theta_{\mathrm{segment}}}\right\rfloor +1$$
where $\theta_{\mathrm{segment}}=\frac{360^{\circ }}{n}=45^{\circ }$ for n=8 points. For the boundary case where $\theta =360^{\circ }$, the line index is adjusted to maintain circular continuity:(40)$$\iota =\left\{\begin{array}{cc} \left\lfloor \frac{\theta }{\theta_{\mathrm{segment}}}\right\rfloor +1 & \mathrm{if}\;\left\lfloor \frac{\theta }{\theta_{\mathrm{segment}}}\right\rfloor +1\leq n\\ 1 & \mathrm{otherwise} \end{array}\right.$$

Once the appropriate line segment is identified, the algorithm extracts the start and end points of the selected polygon edge in 3D space:(41)$$\begin{array}{cc} P_{\mathrm{start}} & =N_{\iota }\left[1:3\right]={\left[P_{\mathrm{start},x},P_{\mathrm{start},y},P_{\mathrm{start},z}\right]}^{T} \end{array}$$(42)$$\begin{array}{cc} P_{\mathrm{end}} & =N_{\iota }\left[4:6\right]={\left[P_{\mathrm{end},x},P_{\mathrm{end},y},P_{\mathrm{end},z}\right]}^{T} \end{array}$$

The ray-line intersection calculation employs parametric equations to find the intersection between the direction ray from the origin and the identified polygon edge. The direction ray is defined by its origin and direction $d^{\left(2\right)}={\left[\cos\left(\theta \right),\sin\left(\theta \right)\right]}^{T}$:(43)$$R\left(t\right)=O^{\left(2\right)}+t\;d^{\left(2\right)}$$
where $O^{\left(2\right)}={\left[0,0\right]}^{T}$ is the origin in 2D space and t≥0 is the parameter along the ray.

The corresponding line segment is described by its endpoints $P_{\mathrm{start}}^{\left(2\right)}={\left[P_{\mathrm{start},x},P_{\mathrm{start},y}\right]}^{T}$ and $P_{\mathrm{end}}^{\left(2\right)}={\left[P_{\mathrm{end},x},P_{\mathrm{end},y}\right]}^{T}$:(44)$$L\left(s\right)=P_{\mathrm{start}}^{\left(2\right)}+s\;\left(P_{\mathrm{end}}^{\left(2\right)}-P_{\mathrm{start}}^{\left(2\right)}\right),\;s\in \left[0,1\right]$$

To identify the intersection, we equate the ray and segment parametric forms:(45)$$O^{\left(2\right)}+t\;d^{\left(2\right)}=P_{\mathrm{start}}^{\left(2\right)}+s\;\left(P_{\mathrm{end}}^{\left(2\right)}-P_{\mathrm{start}}^{\left(2\right)}\right)$$ This yields a linear system for the parameters *t* and *s*:(46)$$\left[\begin{array}{cc} d_{x} & -(P_{\mathrm{end},x}-P_{\mathrm{start},x})\\ d_{y} & -(P_{\mathrm{end},y}-P_{\mathrm{start},y}) \end{array}\right]\left[\begin{array}{c} t\\ s \end{array}\right]=P_{\mathrm{start}}^{\left(2\right)}-O^{\left(2\right)}$$
where the parameter *t* represents the distance along the ray direction from the origin $O^{\left(2\right)}$, while *s* denotes the relative position along the line segment, with s=0 at $P_{\mathrm{start}}^{\left(2\right)}$ and s=1 at $P_{\mathrm{end}}^{\left(2\right)}$. Values of *s* between 0 and 1 correspond to points within the segment, and clamping *s* ensures the intersection point remains within the segment bounds. If the determinant of this matrix is sufficiently small, the ray and segment are nearly parallel, and the midpoint of the segment is selected as a fallback to ensure robustness. Otherwise, we solve for *t* and *s* and clamp *s* to the interval [0,1] to guarantee the intersection point remains within the segment bounds.

To solve for the intersection, the system is addressed for the parametric coefficients, with special handling for the case of nearly parallel lines where |det(A)|<ϵ (with ϵ typically set to $10^{-10}$). In such situations, the midpoint of the line segment is used as a fallback:(47)$$s=\left\{\begin{array}{cc} {\left(A^{-1}b\right)}_{2} & \mathrm{if}\;|\det(A)|\geq \epsilon \\ 0.5 & \mathrm{if}\;|\det(A)|<\epsilon \end{array}\right.$$ To ensure the intersection remains within the segment, the parameter *s* is clamped to the interval [0,1]:(48)$$s_{\mathrm{clamped}}=\max\left(0,\min\left(1,s\right)\right)$$ The corresponding 3D intersection point is then obtained by linear interpolation between the segment endpoints:(49)$$P_{\mathrm{intersection}}=P_{\mathrm{start}}+s_{\mathrm{clamped}}\cdot \left(P_{\mathrm{end}}-P_{\mathrm{start}}\right)$$ The weighted distance is finally computed as the Euclidean distance from the origin to this intersection point:(50)$$q=\sqrt{\sum_{k=1}^{3}{\left(P_{\mathrm{intersection},k}-O_{k}\right)}^{2}}$$

The final output of the proposed method is the pair (θ,q), where θ is the estimated movement artifact direction (in degrees) as defined in Equation (Equation 35) and Section 2.1, and *q* is the weighted artifact magnitude (in pixels) along the principal direction. Together, (θ,q) provide a precise and interpretable quantification of movement artifacts, supporting downstream applications with geometric accuracy and computational efficiency.

## 3. Results

This section presents the results and analysis in the same sequence as outlined in the methodology overview (Figure 2), ensuring clarity and logical progression. First, we report the outcomes of the mathematical directional movement artifact modeling, followed by the data preparation results. Next, we detail the findings from the directional movement artifact quantification preparation. Finally, we present the integrated results of 3D space mapping, optimal triplet selection, direction estimation, and weighted movement artifact quantification calculation, as these components are closely interdependent and best interpreted together.

### 3.1. Signal-Domain Representation of Movement Artifacts Results

This section presents results synthesized from the mathematical model described previously, focusing on the effects of orientation (φ) and velocity (*v*) on movement artifact formation. By systematically varying φ from $0^{\circ }$ to $165^{\circ }$ in $15^{\circ }$ increments, the model demonstrates clear directional movement artifact patterns, supporting the theoretical basis for estimating artifact direction θ, as illustrated in Figure 9.

The visibility of movement artifacts increases with velocity, becoming prominent at v≥0.2 m/s. However, excessively high velocities may produce artifacts that are indistinguishable from valid image structures, potentially exceeding the optimal detection range. In practical imaging scenarios, including medical applications, observable movement artifacts typically occur within the 0 to 0.3 m/s range, as shown in Figure 10.

These synthesized results validate the model’s capacity to predict movement artifact direction θ and support the feasibility of the proposed estimation framework. Nevertheless, empirical validation requires real-world datasets with properties beyond those achievable through simulation, motivating the development of the experimental dataset described in Section 3.2.

### 3.2. Data Preparation Results

The empirical image employed in the proposed methodology was acquired using a precision-controlled electronic apparatus, as described in Section 2.2. The resulting images are systematically organized according to the two principal parameters: orientation (φ) and velocity (*v*). Specifically, columns represent orientation angles from $0^{\circ }$, $15^{\circ }$, $30^{\circ }$, up to $165^{\circ }$, and their complementary angles from $180^{\circ }$, $195^{\circ }$, $210^{\circ }$, up to $345^{\circ }$, arranged left to right. Rows correspond to velocities from 0, 0.1, 0.2, up to 1.0 m/s, arranged top to bottom. For each (φ,v) combination, at least one sample image is acquired and processed, as shown in Figure 11. The empirical image at $30^{\circ }$ orientation and 0.2 m/s velocity (highlighted by the dashed frame) is selected for detailed illustration in Section 3.3.

### 3.3. Directional MAQ Preparation Results

To quantify movement artifacts in pixels, empirical images at each orientation and velocity are processed using the detection and quantification pipeline. While all images undergo this procedure, results for the $30^{\circ }$ orientation and 0.2 m/s velocity are presented in Figure 12 to illustrate the internal algorithmic steps. The figure displays outputs for MAPE, ROPE, and MAQ, arranged left to right for horizontal, diagonal, vertical, and antidiagonal directions. MAPE identifies artifact locations and their complements (red and blue dots), ROPE estimates the reference origin (green dots), and MAQ quantifies the pixel-wise displacement between artifact and origin, suppressing irrelevant data. These results correspond to the original image coordinates and exemplify the pipeline’s operation on a single instance.

When applied across the entire dataset, the MAQ values for each image and direction are aggregated and visualized as a heatmap in Figure 13. The heatmap reveals that the horizontal, diagonal, vertical, and antidiagonal directions exhibit maximal MAQ responses at $0^{\circ }$, $45^{\circ }$, $90^{\circ }$, and $135^{\circ }$, respectively. In each direction, the MAQ value increases with velocity up to approximately 0.7 m/s before declining, reflecting the relationship between artifact magnitude and motion parameters.

### 3.4. Direction Estimation and Weighted Distance Calculation Results

This section presents the experimental outputs of the proposed methodology: the estimated movement artifact direction θ (in degrees) and the weighted distance *q* (in pixels). Figure 14 illustrates representative results for θ and *q* at a velocity of 0.2 m/s across all tested orientations φ ($0^{\circ }$ to $165^{\circ }$ in $15^{\circ }$ increments), with both 3D and 2D visualizations provided. The 2D view corresponds to the top-down projection of the 3D space (excluding the MAQ-reliability axis, which does not affect θ calculation). Ground truth orientation φ is systematically compared to estimated direction θ using angular error $AE=\min(|\theta_{est}-\varphi_{gt}|,360^{\circ }-|\theta_{est}-\varphi_{gt}|)$ and velocity-dependent analysis across v∈[0.0,1.0] m/s, demonstrating that estimated directions closely align with true orientations and confirming the effectiveness of the proposed approach. Figure 14 provides a comprehensive visual summary, while Table 1 reports the detailed estimated directions for all orientations and velocities in the dataset.

Table 1 summarizes the estimated movement artifact direction θ (degrees) across velocities *v* and ground-truth orientations φ. The tabulated values reveal a clear velocity dependence: estimation errors are notably higher at very low velocities (v≤0.1 m/s), where artifacts are present but lack sufficient separability to expose a dominant orientation. As *v* increases beyond 0.2 m/s, directional cues become more pronounced and estimates stabilize across φ, yielding consistently low angular deviations. This behavior is consistent with the bidirectional nature of movement artifacts, where energy may manifest along both the primary and complementary angles; consequently, for certain (v,φ) pairs, estimates may fall within either 0–$180^{\circ }$ or 180–$360^{\circ }$ while still indicating the correct orientation class. For clarity, Figure 15 visualizes the error structure over (v,φ), complementing the trends in the table.

Figure 15 presents the angular error distribution as a function of orientation and velocity. Figure 15a shows that errors peak at low velocities, particularly around $30^{\circ }$ at v=0.1 m/s, and then decrease markedly as *v* increases; for example, the error drops to approximately $7^{\circ }$ near $30^{\circ }$ at v=0.6 m/s. Beyond v≈0.2 m/s, the error surface flattens across orientations, indicating stable estimation in the moderate-velocity regime. Figure 15b highlights the angular structure from $0^{\circ }$ to $360^{\circ }$, with the lowest errors concentrated along the principal axes ($0^{\circ }$, $45^{\circ }$, $90^{\circ }$, $135^{\circ }$). Error symmetry about complementary angles reflects the intrinsic bidirectionality; occasional $180^{\circ }$ flips can occur in low-SNR regimes but become rare once v≥0.2 m/s.

Confining analysis to the four principal axes provides a favorable accuracy–efficiency trade-off for direction estimation. As emphasized in Section 2.4.1, these axes capture the dominant error minima, enabling substantial computational savings (approximately a 45× reduction) while maintaining marginal angular deviation on the order of $\pm 5^{\circ }$ under typical conditions. Practically, an operational threshold emerges near v≥0.1 m/s; once v>0.2 m/s, directional estimation becomes consistently high-performance and remains stable at greater velocities. The often-cited 0.7 m/s boundary pertains to a separate magnitude-oriented consideration discussed subsequently and does not impose an upper limit on the reliability of the direction estimates.

In the following context, we turn to the weighted movement artifact magnitude *q* (pixels) to characterize where the upper-velocity constraint arises and how it informs practical operation.

Table 2 reports the estimated weighted movement artifact magnitude *q* (in pixels) as a function of velocity *v* across orientations. These values are magnitude estimates—not errors—and they exhibit a clear velocity-driven trend: *q* remains small at low velocities, increases steadily with *v*, and approaches its maximum near v≈0.6 m/s. This behavior is expected, since lower motion naturally yields smaller artifact magnitudes; consequently, there is no lower-bound limitation for *q*—small values at low *v* are valid and informative, rather than a failure mode. The visualizations in Figure 16 are consistent with the tabulated results: Figure 16a (Cartesian coordinates) highlights the growth of *q* with increasing *v* followed by a gentle roll-off, while Figure 16b (polar coordinates) shows radially expanding magnitudes with largely symmetric behavior over orientations. The only practical constraint for interpreting *q* arises beyond approximately 0.6 m/s, where magnitude responses become ambiguous due to saturation and overlap of artifact structures with image content, leading to reduced separability. Therefore, in the context of weighted distance estimation, the upper-velocity regime defines the sole limitation, whereas low-velocity regimes simply reflect proportionally small but valid *q* values. These findings provide a pixel-scale, quantitatively interpretable measure of movement artifact strength that is directly actionable for downstream compensation and image quality assessment.

In practical applications, the overall system performance is determined by jointly considering the limitations of both movement artifact direction estimation and weighted distance quantification, which are directly influenced by changes in velocity *v*. This relationship is visualized in Figure 17. Figure 17a,b display the system’s capabilities in Cartesian and polar coordinates, respectively. The contour maps in these subfigures represent the average sum of normalized direction errors and normalized weighted distances, with values ranging from 0 (dark blue, indicating optimal performance) to 1 (dark red, indicating boundary limitation).

The dark blue regions correspond to velocity ranges where there is sufficient information to accurately estimate both the direction and magnitude of movement artifacts. In contrast, dark red regions indicate performance degradation—either due to insufficient data at low velocities (making direction estimation unreliable) or ambiguity at high velocities (where weighted distance quantification becomes indistinct and artifacts may be confused with inherent image features). The continuous high-performance zone, spanning velocities from approximately 0.1 to 0.7 m/s (dotted to dashed lines), defines the operational “safe zone” for the proposed method. This range encompasses typical velocities encountered in practical imaging scenarios, where acceptable motion artifacts are generally below 0.3 m/s.

Figure 17c presents a quadratic polynomial regression analysis of the normalized errors. The blue triangles and line represent the normalized average direction estimation error and its trend, with shaded areas indicating standard deviation across velocity ranges. The lower bound of reliable performance is observed near 0.1 m/s (dotted line). The red line shows the normalized average weighted distance, with an upper bound of approximately 0.7 m/s (dashed line).

Optimal system performance is achieved when the blue line (direction error) is minimized and the red line (weighted distance) is maximized, confirming that the most reliable operation occurs within the 0.1–0.7 m/s velocity range.

The computational complexity analysis presented in Table 3 highlights the practical efficiency of the proposed signal processing framework relative to conventional movement artifact direction estimation methods. Deep learning and multi-frame approaches incur significantly greater computational costs ($O(N^{3})$ and $O(N^{2}K)$, respectively) and resource requirements, limiting their suitability for real-time and single-frame applications. In contrast, the proposed pipeline—comprising MAPE, ROPE, MAQ, and the MADE algorithm—achieves comprehensive multi-directional estimation with an overall complexity of $O(N^{2}4)$ and minimal demands on memory, CPU, and GPU resources, as further detailed in Table 4.

Furthermore, the MADE algorithm leverages 3D geometric analysis to deliver precise direction estimation with negligible additional overhead, while the entire pipeline remains highly efficient and suitable for real-time processing. The lightweight nature of the proposed method further ensures its applicability on single-board computers (e.g., Raspberry Pi), Microcontrollers (e.g., ESP32) and system-on-chip (SoC) platforms, supporting practical integration in embedded and portable imaging systems. The results confirm that the proposed method not only matches or exceeds the directional accuracy of existing techniques but also offers significant improvements in efficiency, scalability, and single-frame capability. These findings substantiate the theoretical and practical contributions of the signal processing approach, positioning it as a robust alternative to deep learning and frequency domain methods for motion blur analysis, especially movement artifact direction estimation.

## 4. Discussion

The proposed method leverages signal processing principles to enable real-time movement artifact direction estimation with minimal computational resources, contrasting sharply with deep learning and multi-frame methods that demand substantial hardware and computational complexity. The pipeline’s simplicity allows deployment on conventional embedded systems (Raspberry Pi, ESP32, system-on-chip platforms) requiring only 10 MB RAM and a single CPU core, making it ideal for resource-constrained and portable imaging applications.

The framework is grounded in discrete signal processing, utilizing spatial self-similarity analysis, correlation, and related operations. While the core processes are straightforward, practical implementations may benefit from fast algorithms, such as the Fast Fourier Transform, to further reduce computational complexity from $O(N^{2})$ to O(NlogN) in certain sub-components. Additionally, the pipeline is amenable to parallelization; for example, the Directional MAQ Preparation step can be distributed across GPU cores, as each image row is processed independently, potentially reducing processing time by a factor of *N*.

Although the primary focus is on movement artifact direction estimation across the full angular range ($0^{\circ }$–$360^{\circ }$), direct comparison of estimation errors with other methods is challenging due to differences in task objectives, accuracy metrics, and processing speed priorities. The diversity of approaches—ranging from deep learning and multi-frame analysis to frequency domain and conventional computer vision—further complicates fair benchmarking. This work emphasizes a theoretically fundamental signal processing approach, enabling robust operation on real-world data and resource-limited hardware.

Several limitations merit consideration for practical deployment. The methodology demonstrates robust performance under controlled conditions but faces challenges including illumination variability, where intensity variations may be misinterpreted as movement artifacts, and noise sensitivity in low-quality images, despite EMA noise suppression in ROPE. Complex background scenarios with textured backgrounds or multiple objects may introduce false positives, while the motion pattern assumptions of linear motion with oscillatory components may limit accuracy for irregular movement patterns. The dataset employed features velocity increments of 0.1 m/s, which may obscure subtle phenomena at low velocities and impact directional accuracy. Finer sampling intervals and denser orientation steps (e.g., reducing from 15° to 10°, 5°, or 1°) could enhance resolution but require extensive data preparation. While the current pipeline focuses on four principal directions, adaptive selection of additional axes (e.g., 8, 16, or 32) may further improve accuracy, albeit with increased computational demands. Expanding to more complex patterns, including irregular object geometries, textured surfaces, multi-object scenarios, and realistic medical imaging conditions with tissue heterogeneity, could reveal new insights into algorithm robustness and applicability. Future validation should include varying noise levels (Gaussian, salt-and-pepper, and speckle), illumination conditions (non-uniform lighting, shadows, and contrast variations), motion complexity (non-linear trajectories, acceleration patterns), and adaptive parameter selection for enhanced robustness across diverse imaging modalities and clinical scenarios.

Future research will focus on integrating the estimated direction and magnitude parameters with Modified-DFT-based compensation techniques to evaluate overall artifact correction performance. The methodology will then be extended to real-world imaging applications, particularly the LUIAS, where movement artifacts significantly impact image quality assessment. By enabling accurate artifact characterization and compensation, this work aims to enhance surveillance capabilities and improve detection accuracy for high-risk patients, supporting timely identification of abnormalities by healthcare professionals.

## 5. Conclusions

This work presents a signal processing-based framework for movement artifact direction estimation in single-frame images, addressing key limitations of existing deep learning, frequency domain, and multi-frame approaches. By integrating multi-directional preprocessing algorithms (MAPE, ROPE, MAQ) with the MADE algorithm’s 3D geometric analysis, the proposed method achieves comprehensive estimation of two principal outputs: the direction of movement artifacts (in degrees) and their weighted quantity (in pixels). These outputs provide both precise angular information and quantitative assessment of artifact magnitude, supporting interpretable and actionable results for downstream applications.

Experimental results demonstrate robust performance and instantaneous processing capability under typical computing environments. Theoretically, the approach offers a fundamental spatial domain solution for accurate movement artifact quantification and direction estimation. Practically, it demonstrates potential applicability in domains such as medical imaging, surveillance, and embedded systems. Limitations include dataset granularity and the scope of tested image shapes, which may be addressed in future work by expanding sampling intervals and exploring adaptive multi-directional analysis. Future research will focus on applying the methodology to real-world movement artifact compensation, particularly in the LUIAS, to enhance image quality and surveillance capabilities.

In summary, the proposed framework delivers efficient and scalable movement artifact direction estimation, with explicit emphasis on both directional accuracy and quantitative measurement, advancing the theoretical and practical aspects of signal processing and image analysis.

## Figures and Tables

**Figure 1 sensors-25-07487-f001:**
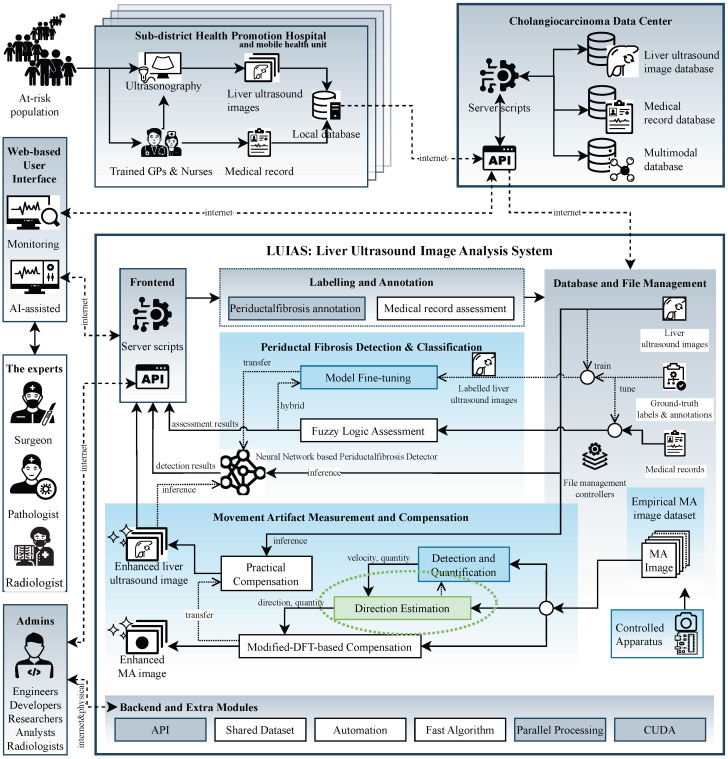
Overview of the Liver Ultrasound Image Analysis System (LUIAS) in the Cholangiocarcinoma Surveillance System.

**Figure 2 sensors-25-07487-f002:**
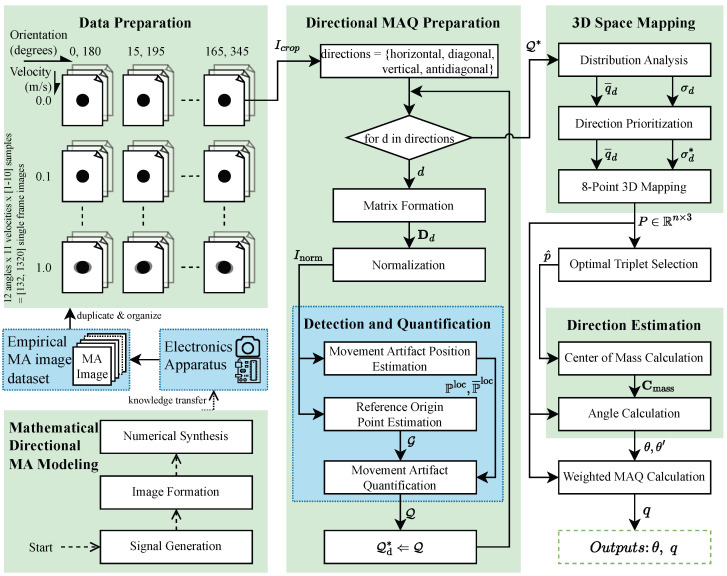
Overview of the proposed Movement Artifact Direction Estimation methodology.

**Figure 3 sensors-25-07487-f003:**
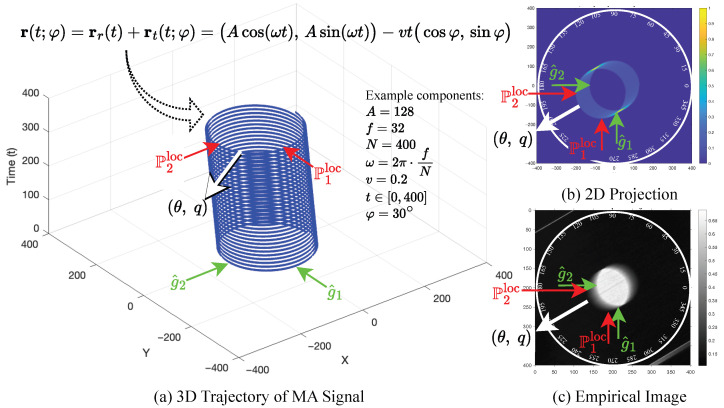
Mathematical modeling of movement artifact signal. Each subfigure contains five arrows pointing to the same corresponding areas: green arrows (${\hat{g}}_{1}$ and ${\hat{g}}_{2}$) indicate some origin points, red arrows ($P_{1}^{\mathrm{loc}}$ and $P_{2}^{\mathrm{loc}}$) indicate some movement artifact points, and white arrows highlight the most important parameters targeted by this paper, $\left(\theta ,\;q\right)$, estimated direction in degrees and distance in pixels.

**Figure 4 sensors-25-07487-f004:**
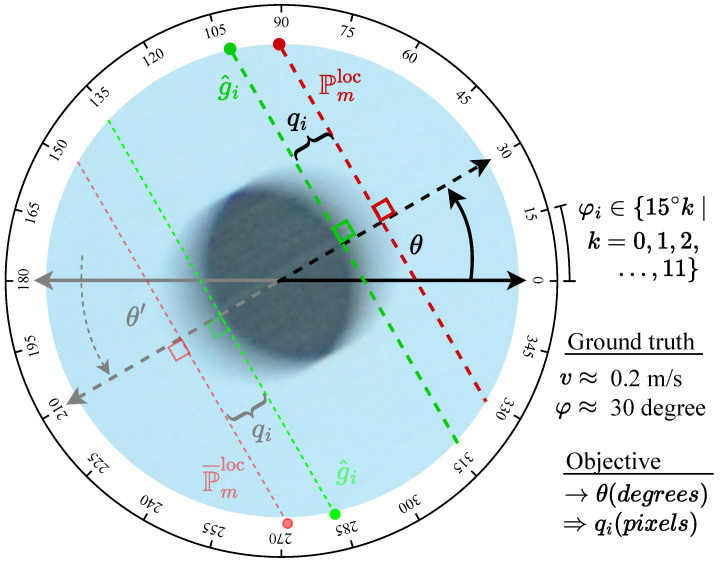
Unified object movement and principal parameters for movement artifact estimation.

**Figure 5 sensors-25-07487-f005:**
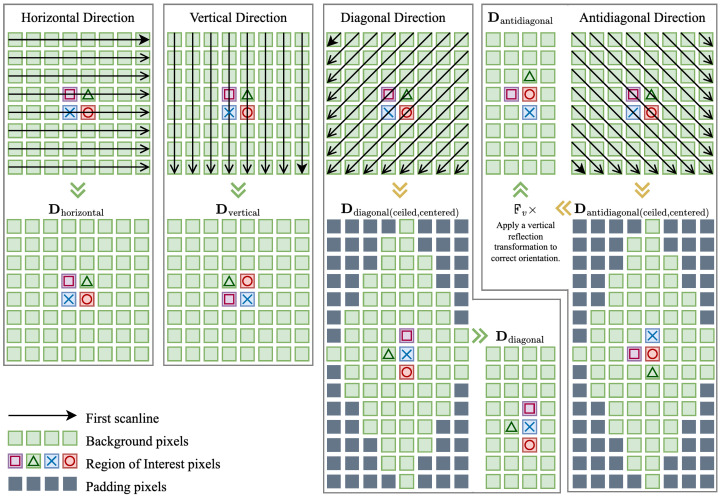
Matrix formation process for directional MAQ preparation.

**Figure 6 sensors-25-07487-f006:**
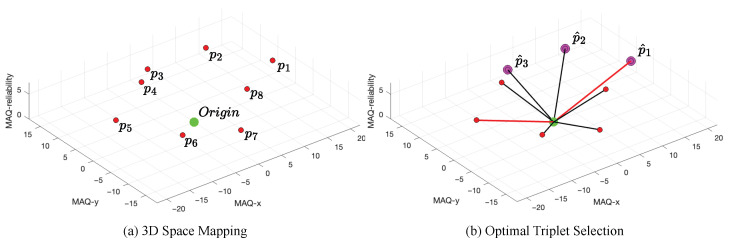
(**a**) Three-dimensional space mapping: Directional MAQ means are represented as points $p_{1}$ to $p_{8}$ in 3D space (MAQ-x, MAQ-y, MAQ-reliability), each correlated to the origin (green, “MAQ Origin”). (**b**) Optimal triplet selection: The three most distant adjacent points (${\hat{p}}_{1}$, ${\hat{p}}_{2}$, ${\hat{p}}_{3}$) are highlighted, with red lines indicating the maximal distances included in the selected triplet.

**Figure 7 sensors-25-07487-f007:**
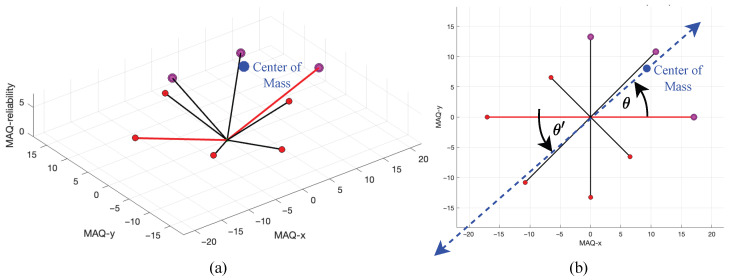
Illustration of center of mass calculation and its geometric interpretation, (**a**) 3D view, (**b**) 2D view.

**Figure 8 sensors-25-07487-f008:**
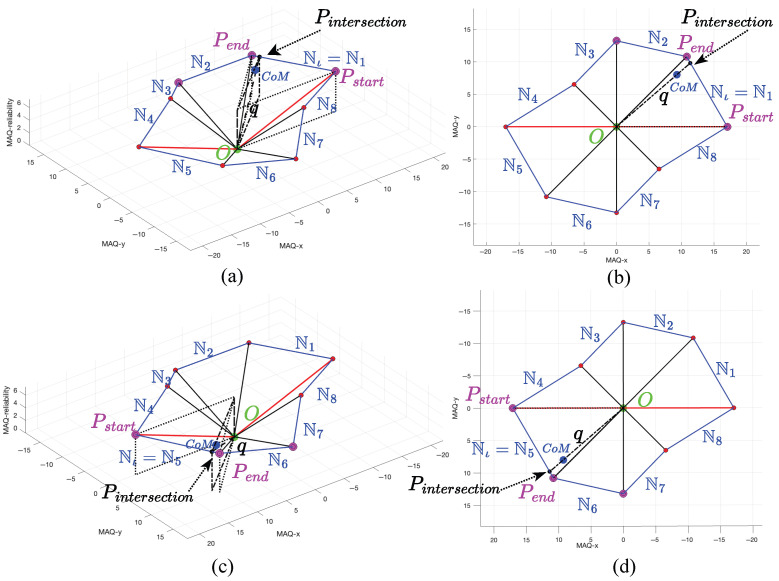
Illustration of the weighted distance calculation process. Subfigures (**a**,**b**) present the 3D and 2D views of the calculation along the corresponding line segment, while (**c**,**d**) depict the complementary relationships, respectively.

**Figure 9 sensors-25-07487-f009:**
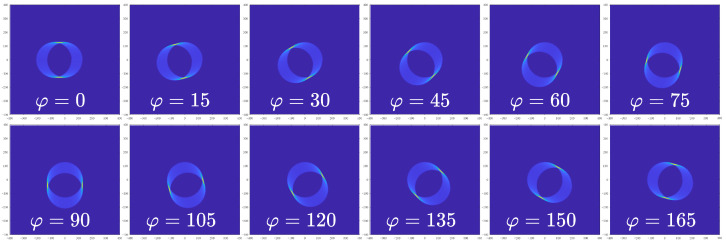
Synthesized movement artifact patterns for varying orientation angles φ with v=0.2 m/s.

**Figure 10 sensors-25-07487-f010:**
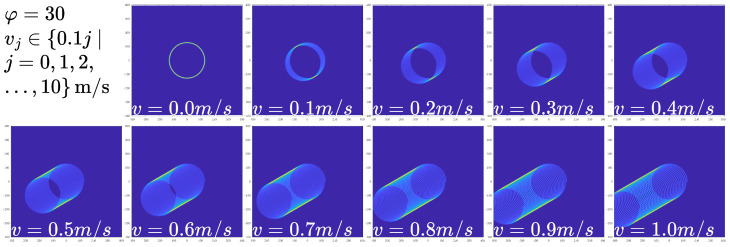
Synthesized movement artifact patterns for varying velocities *v* with $\varphi =30^{\circ }$.

**Figure 11 sensors-25-07487-f011:**
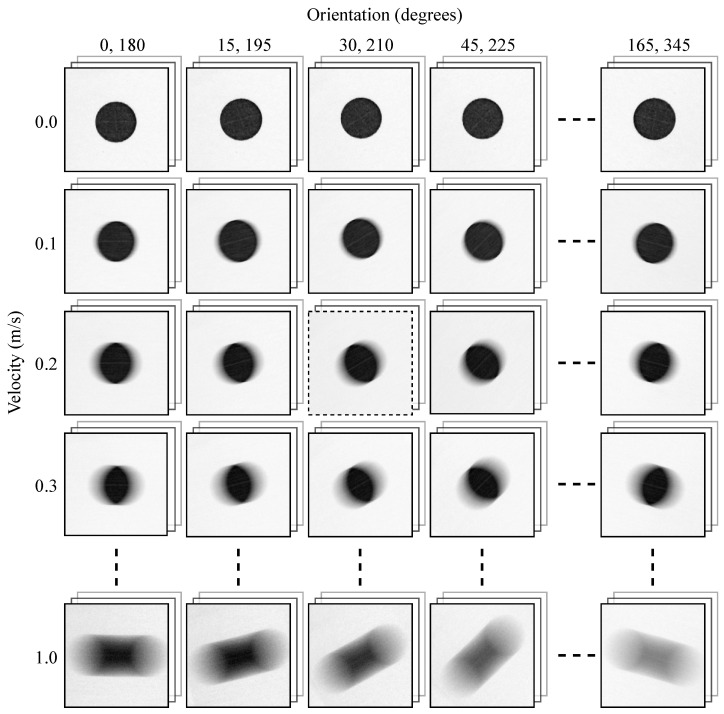
Empirical images for all orientations and velocities in the dataset.

**Figure 12 sensors-25-07487-f012:**
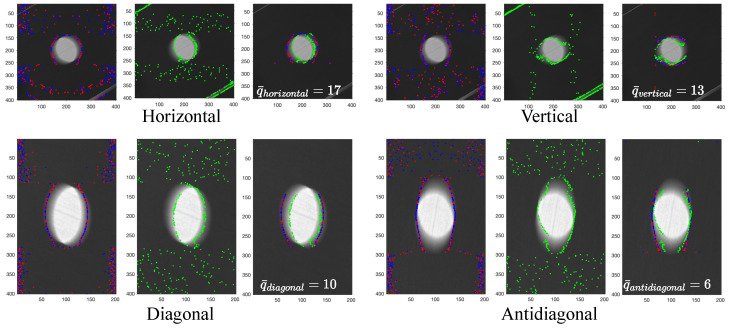
Directional MAQ preparation results for the empirical image at $30^{\circ }$ orientation and 0.2 m/s velocity.

**Figure 13 sensors-25-07487-f013:**
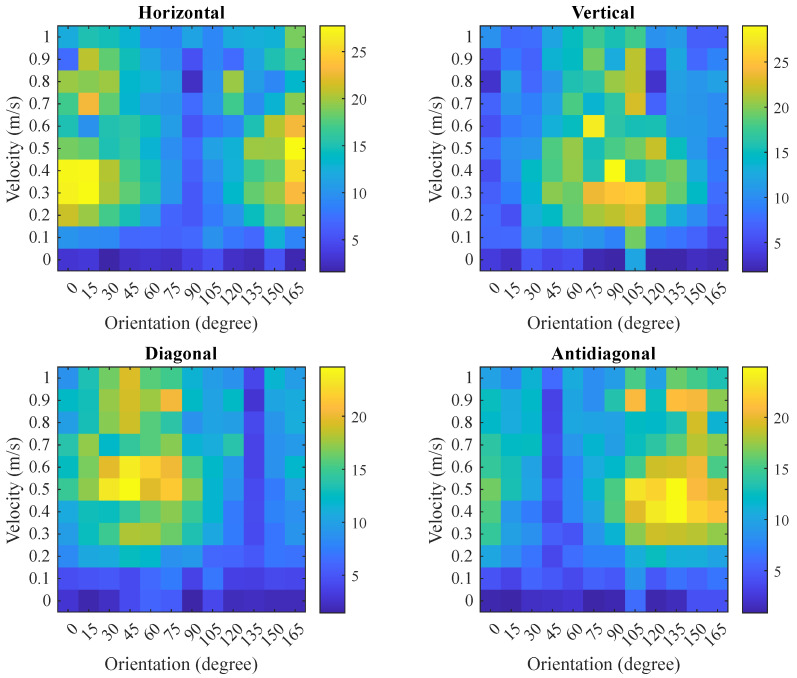
Directional MAQ preparation results heatmap for all orientations and velocities. Color bar indicates MAQ value in pixels.

**Figure 14 sensors-25-07487-f014:**
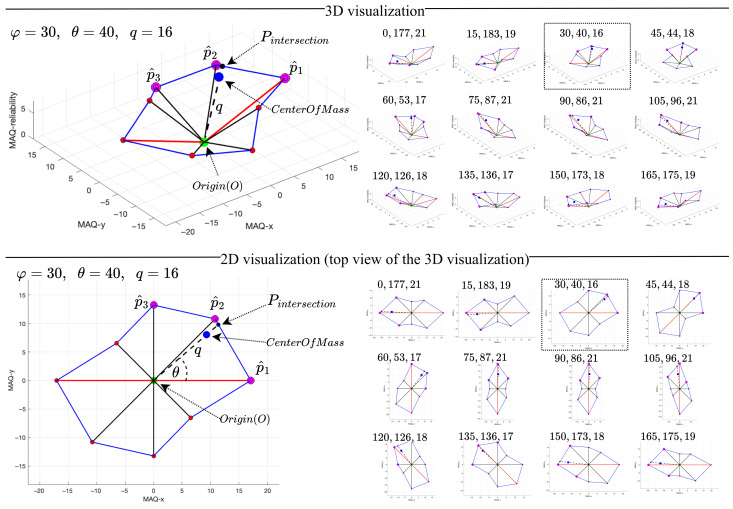
Visualization of the integrated results of the proposed MADE algorithm for the empirical image at 0.2 m/s velocity.

**Figure 15 sensors-25-07487-f015:**
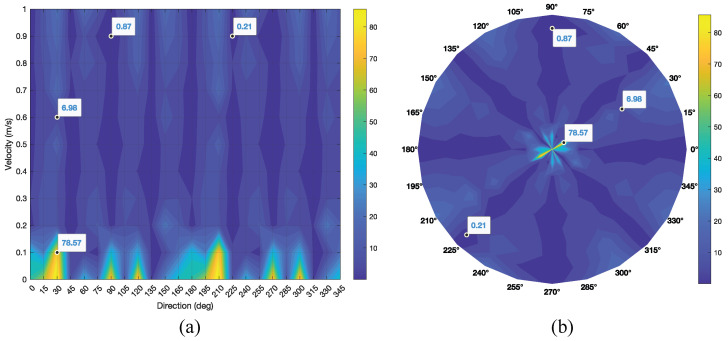
Estimation error for each velocity and orientation, (**a**) Cartesian coordinates, (**b**) polar coordinates.

**Figure 16 sensors-25-07487-f016:**
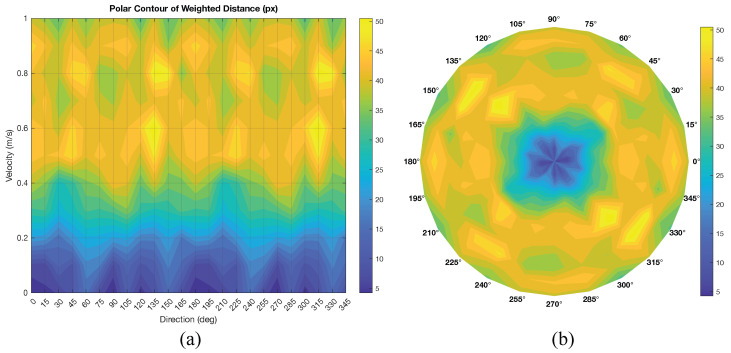
Plottings of estimated weighted movement artifact distance (*q*) in pixels, (**a**) Cartesian coordinates; (**b**) polar coordinates.

**Figure 17 sensors-25-07487-f017:**
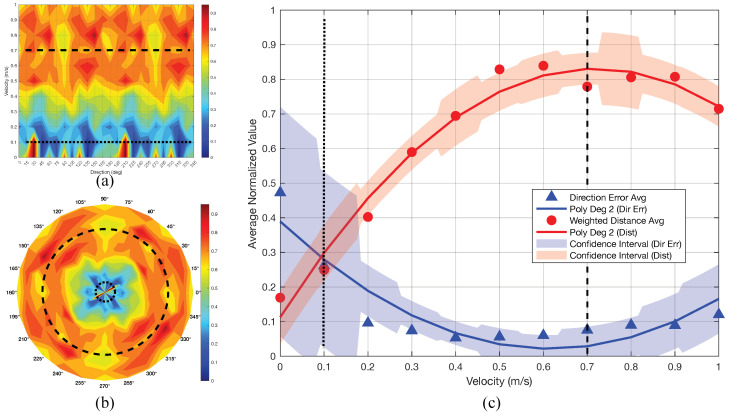
Comprehensive performance evaluation of the proposed methodology. (**a**) Integrated performance analysis for direction estimation and weighted distance quantification, presented in Cartesian coordinates. (**b**) Corresponding performance visualization in polar coordinates. (**c**) Polynomial regression analysis of performance metrics.

**Table 1 sensors-25-07487-t001:** Estimated movement artifact direction (θ) in degrees for each velocity (*v*) in m/s and orientation (φ) in degrees.

φ	0.0	0.1	0.2	0.3	0.4	0.5	0.6	0.7	0.8	0.9	1.0
$0^{\circ }/180^{\circ }$	48	139	178	176	177	178	178	182	178	179	180
$15^{\circ }/195^{\circ }$	132	40	183	184	186	187	188	188	186	184	184
$30^{\circ }/210^{\circ }$	120	131	39	39	39	195	37	190	195	190	187
$45^{\circ }/225^{\circ }$	54	51	45	47	45	45	44	48	41	45	45
$60^{\circ }/240^{\circ }$	202	54	52	52	55	54	54	53	49	51	80
$75^{\circ }/255^{\circ }$	55	77	88	85	82	79	81	83	83	78	85
$90^{\circ }/270^{\circ }$	193	130	88	89	92	89	89	89	92	89	89
$105^{\circ }/285^{\circ }$	95	123	96	95	98	100	97	98	99	99	95
$120^{\circ }/300^{\circ }$	40	84	127	128	124	127	128	129	104	102	99
$135^{\circ }/315^{\circ }$	132	133	139	134	135	136	135	132	134	135	135
$150^{\circ }/330^{\circ }$	121	145	174	142	144	144	163	141	138	166	171
$165^{\circ }/345^{\circ }$	121	143	152	174	169	170	170	170	174	177	176

**Table 2 sensors-25-07487-t002:** Estimated weighted movement artifact distance (*q*) in pixels for each velocity (*v*) in m/s and direction (θ) in degrees.

θ	0.0	0.1	0.2	0.3	0.4	0.5	0.6	0.7	0.8	0.9	1.0
$0^{\circ }/180^{\circ }$	4	8	22	31	40	45	46	40	44	47	40
$15^{\circ }/195^{\circ }$	6	8	21	32	37	43	43	44	40	44	38
$30^{\circ }/210^{\circ }$	7	14	17	26	29	44	40	37	39	39	34
$45^{\circ }/225^{\circ }$	10	11	21	29	31	48	47	39	47	44	40
$60^{\circ }/240^{\circ }$	17	20	25	32	36	42	43	43	48	40	36
$75^{\circ }/255^{\circ }$	8	18	24	32	37	45	42	40	37	43	40
$90^{\circ }/270^{\circ }$	4	8	23	33	43	42	42	38	38	44	41
$105^{\circ }/285^{\circ }$	12	15	23	36	42	45	43	43	41	45	40
$120^{\circ }/300^{\circ }$	5	10	19	31	35	41	46	40	39	40	33
$135^{\circ }/315^{\circ }$	9	12	19	29	41	49	53	42	51	44	41
$150^{\circ }/330^{\circ }$	15	19	23	30	35	39	43	42	50	37	34
$165^{\circ }/345^{\circ }$	9	15	18	33	35	43	43	44	36	44	37

**Table 3 sensors-25-07487-t003:** Computational complexity analysis per image for major movement artifact direction estimation methods and proposed algorithm components.

Method	Complexity	Description
Deep Learning	$O(N^{3})$	Multiple convolutional layers, batch processing, high computational overhead
Multi-frame	$O(N^{2}F)$	Multiple frames processed dependently, higher temporal coherence, where F is the number of frames
Iterative	$O(N^{2}T)$	Iterative signal analysis, repeated until optimal result is achieved, where T is the number of iterations
**Proposed Signal Processing**	$O(N^{2}4)$	Efficient for single-frame, covers all directional outputs by operating only the necessary 4 directions
MAPE	$O(N^{2})$	Self-similarity analysis and peak detection per scanline
ROPE	O(N)	Exponential moving average and slope analysis per scanline
MAQ	O(N)	Adaptive filtering and quantification for each direction
MADE	O(4)	3D mapping, triplet selection, geometric analysis, 4 directions

**Table 4 sensors-25-07487-t004:** System resource usage comparison for major movement artifact direction estimation methods and proposed algorithm components.

Method	Memory Usage	CPU Usage	GPU Usage	Single-Frame	Real-Time Capability
Deep Learning	High	High	High	Yes	No
Multi-frame	Moderate	Moderate	Moderate	No	Limited
Iterative	Moderate	High	Low	Yes	Yes
**Proposed Signal Processing**	Low	Low	Low	Yes	Yes

## Data Availability

The dataset and code that support the findings of this study will be published publicly on GitHub at https://github.com/nonsakhoo (accessed on 2 December 2025). Until then, they are available from the corresponding author upon reasonable request.

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
