# Peer review of "Movement Artifact Direction Estimation Based on Signal Processing Analysis of Single-Frame Images"

_sensors, 2025, doi:10.3390/s25247487_

Round 1
Reviewer 1 Report
Comments and Suggestions for Authors
This manuscript presents a signal-processing-based framework named Movement Artifact Direction Estimation (MADE) for estimating the direction and magnitude of motion artifacts in single-frame images.The paper is well-structured, technically detailed, and demonstrates solid theoretical grounding. The mathematical modeling and empirical validation are comprehensive. However, several minor revisions are required before acceptance.
1. Some sections (e.g., Sections 2.3–2.4) are overly verbose and could benefit from concise summarization. Consider condensing equations that are purely illustrative into appendices to improve readability.
2. Clarify the evaluation metric used for “alignment to ground truth parameters” in Section 3.
3. Include quantitative comparison (e.g., error in direction estimation) between MADE and existing frequency-domain or CNN-based methods to substantiate claims of efficiency and accuracy.
4. Expand the discussion of limitations (e.g., robustness under varying illumination or noise) and potential integration with real-time imaging hardware.
Reviewer 2 Report
Comments and Suggestions for Authors
I conclude that this study does not have the practical significance stated by the authors and its reliability is questionable. Therefore, I propose to reject this manuscript.

Reviewer 3 Report
Comments and Suggestions for Authors
Further exploration to improve robustness in the low-SNR regime can be added.
Reviewer 4 Report
Comments and Suggestions for Authors
The article is well written with the necessary theoretical background and discussion. The author's effort is commendable. The following may be considered to improve:
- The paper's length (32) and citation (70) are a minor concern. It may be reduced to enhance the reader's interest.
- While the figures are well presented, some axes have small fonts, and they also show all minor grid labels.
- Some of the definitions are defined many times (for ex. MAQ, MAPE, etc), some of them are not defined (for ex. MRI, MA (defined in the end)
- Figure 1 and related material can be removed, as it was not well discussed in the later sections.
- Citation of unpublished work may be reviewed
Round 2
Reviewer 2 Report
Comments and Suggestions for Authors
The authors' response depresses me and aggravates my attitude towards this study. In my country, ultrasound diagnostics and all manipulations with ultrasound devices are allowed only to senior medical personnel who have received special education, i.e. only certified specialists do this. No one can make any diagnoses based on a single static image. Most often, no conclusions can be drawn from one angle (projection). We need several different angles. And you need to know how they are related to each other. I.e. the conclusion that the doctor makes is the result of an interactive process that is related to what he sees on the monitor and what he does with his hands (how he manipulates the probe).
If the authors had not stated the connection of their research with the LUIAS project, but had written that they were proposing a method for determining and classifying motion artifacts in single images obtained using a video camera (video cameras and nothing else, I categorically insist on this), then I would not have any complaints. The authors have not proved the applicability of these methods for the analysis of ultrasound images. Moreover, I am confident in their inapplicability. The authors did not demonstrate any ultrasound images. The authors have not demonstrated what they call a motion artifact looks like on an ultrasound image. I doubt that motion artifacts are the reason for the large amount of false-positive data. Rather, this is due to poorly chosen probe angles (projections), as the personnel receiving the images are poorly qualified.
The authors categorically do not understand that I do not require mathematical modeling of the formation of an ultrasonic beam. The authors should simulate the imaging process during ultrasound examination. But they persistently experimentally and theoretically simulate image formation in a stationary video camera. I'll try to explain the problem so that the authors understand. Imagine that you are using a high-speed video camera, and the object is moving slowly. Then you won't see the object moving in the photo. Imagine that in your case (with your camera and mover) you got an image of Fig.3c. Imagine that then you printed it out and took it on camera without moving the printed sheet. Your method will not be able to distinguish between these two images. That is, in fact (in your proposed method) you cannot understand whether the object is moving or not.
You should have taken an ultrasound machine and an aquarium with water (preferably with glycerin or ultrasonic gel). Secure the probe shallowly from the surface of the water. Move the soft silicone object under the probe in different directions (along and across the scanning plane). You will be surprised by what you see!
Conclusion: I strongly object to the publication of this study in this context.
